# The Nitro Group Reshapes the Effects of Pyrido[3,4-*g*]quinazoline Derivatives on DYRK/CLK Activity and RNA Splicing in Glioblastoma Cells

**DOI:** 10.3390/cancers16040834

**Published:** 2024-02-19

**Authors:** Sophia S. Borisevich, Tatiana E. Aksinina, Margarita G. Ilyina, Victoria O. Shender, Ksenia S. Anufrieva, Georgij P. Arapidi, Nadezhda V. Antipova, Fabrice Anizon, Yannick J. Esvan, Francis Giraud, Victor V. Tatarskiy, Pascale Moreau, Mikhail I. Shakhparonov, Marat S. Pavlyukov, Alexander A. Shtil

**Affiliations:** 1Ufa Institute of Chemistry, Ufa Federal Research Center of the Russian Academy of Sciences, Ufa 450054, Russiamargarita.kondrova@yandex.ru (M.G.I.); 2Institute of Cyber Intelligence Systems, National Research Nuclear University MEPhI, Moscow 115409, Russia; 3Shemyakin and Ovchinnikov Institute of Bioorganic Chemistry, Russian Academy of Sciences, Moscow 117997, Russia; t.e.aksinina@gmail.com (T.E.A.); nadine.antipova@gmail.com (N.V.A.);; 4Lopukhin Federal Research and Clinical Center of Physical-Chemical Medicine of the Federal Medical and Biological Agency, Moscow 119435, Russia; 5Université Clermont Auvergne, Centre National de la Recherche Scientifique, Clermont Auvergne Institut National Polytechnique, Institute of Chemistry of Clermont-Ferrand, F-63000 Clermont-Ferrand, France; 6Institute of Gene Biology, Russian Academy of Sciences, Moscow 119334, Russia; 7Blokhin National Medical Research Center of Oncology, Moscow 115522, Russia; 8Department of Chemistry, Moscow State University, Moscow 119234, Russia

**Keywords:** pyridoquinazolines, DYRK protein kinases, CLK protein kinases, quantum chemical calculations, gene transcription, RNA splicing, glioblastoma, antitumor drug design

## Abstract

**Simple Summary:**

The complex mode of gene expression regulation is a key reason underlying the biological heterogeneity and clinical diversity of malignant gliomas. In particular, high variabilities of gene expression are associated with alternative RNA splicing, a mechanism that generates the transcripts of different structures and functions from the same gene. Protein kinases of the DYRK and CLK families are part of the splicing regulation machinery; therefore, pharmacological targeting of these enzymes in gliomas is therapeutically relevant. We demonstrate that the pyrido[3,4-*g*]quinazoline scaffold is a source of compounds with differential inhibitory efficacy against individual DYRK and CLK enzymes. Our in silico calculations and omics experiments showed that a single chemical substitution in the scaffold can change the kinase inhibitory profile. This modification yielded the splicing antagonist with a high cytotoxic potency against patient-derived glioma cells.

**Abstract:**

Serine-threonine protein kinases of the DYRK and CLK families regulate a variety of vital cellular functions. In particular, these enzymes phosphorylate proteins involved in pre-mRNA splicing. Targeting splicing with pharmacological DYRK/CLK inhibitors emerged as a promising anticancer strategy. Investigation of the pyrido[3,4-*g*]quinazoline scaffold led to the discovery of DYRK/CLK binders with differential potency against individual enzyme isoforms. Exploring the structure–activity relationship within this chemotype, we demonstrated that two structurally close compounds, pyrido[3,4-*g*]quinazoline-2,10-diamine **1** and 10-nitro pyrido[3,4-*g*]quinazoline-2-amine **2**, differentially inhibited DYRK1-4 and CLK1-3 protein kinases in vitro. Unlike compound **1**, compound **2** efficiently inhibited DYRK3 and CLK4 isoenzymes at nanomolar concentrations. Quantum chemical calculations, docking and molecular dynamic simulations of complexes of **1** and **2** with DYRK3 and CLK4 identified a dramatic difference in electron donor-acceptor properties critical for preferential interaction of **2** with these targets. Subsequent transcriptome and proteome analyses of patient-derived glioblastoma (GBM) neurospheres treated with **2** revealed that this compound impaired CLK4 interactions with spliceosomal proteins, thereby altering RNA splicing. Importantly, **2** affected the genes that perform critical functions for cancer cells including DNA damage response, p53 signaling and transcription. Altogether, these results provide a mechanistic basis for the therapeutic efficacy of **2** previously demonstrated in in vivo GBM models.

## 1. Introduction

Dual specificity tyrosine-phosphorylation-regulated kinases (DYRK1A, -B; class 1; DYRK2-4, class 2) and Сdc-2-like kinases (CLK 1-4) are the families of serine/threonine-tyrosine protein kinases that regulate gene expression via trans-phosphorylation of serine-arginine-rich (SR) proteins, which govern RNA splice site selection. Consequently, an aberrant activation of DYRKs and CLKs can imbalance the maturation of primary transcripts. DYRKs and CLKs have been implicated in the pathogenesis of neurological disorders such as Down syndrome and Parkinson and Alzheimer diseases; in protozoan and viral infections; in metabolic diseases including diabetes, abnormal folate/methionine turnover and bone pathologies (reviewed in [1,2]). Of special importance is the mechanistic role of DYRKs and CLKs in tumor biology. Recent studies have demonstrated that altered RNA splicing is a general hallmark of malignant transformation. The expanding list of tumor types in which deregulated splicing has been mechanistically implicated includes solid cancers of various tissue origins as well as leukemias [3]. Therefore, the design of DYRK/CLK inhibitors with therapeutic properties in cancer and beyond emerges as an intensely investigated area [2,4,5,6].

A major problem in the development of DYRK/CLK inhibitors is their selectivity to individual family members. The diversity within the CLK family arises from the ancestor gene duplication; CLK1/CLK4 share up to 78% homology of amino acid sequences [7]. Similarly, the degree of primary structure identity within the DYRK family is high for both catalytic and activation loops [8], which complicates the development of isoenzyme-specific inhibitors. Nevertheless, substantial interest in these targets produced a series of investigations leading to the identification of inhibitors of various chemical structures [1,9]. For example, DYRK/CLK inhibitors based on the pyrido[3,4-*g*]quinazoline scaffold have been rationally designed based on meridianin derivatives [10] and exhibited nanomolar potencies toward DYRK/CLK protein kinases, with selectivity depending on the pyrido[3,4-*g*]quinazoline substitution pattern [11]. 

Based on molecular modeling analysis of the tentative mode of interaction of the pyrido[3,4-*g*]quinazoline scaffold with the ATP-binding site in DYRK1A, the second generation of DYRK/CLK inhibitors has been produced, aiming at restricting the conformation between the aminopyrimidine and indole moieties (Figure 1, top panel, blue color) and the removal of the NH pyrrole group that was not directly involved in the interactions with the target (green). Also, an additional N atom was introduced to strengthen the H-bond network with the target (Figure 1, top panel, red color). These modifications yielded compounds **1** and **2** (Figure 1, bottom panel), which share the same scaffold but vary in one single function at position 10, namely, -NH_2_ (**1**) vs. -NO_2_ (**2**). The analysis of structurally close derivatives is pertinent to the rational design of isoform-specific inhibitors within the enzymatic families. 

Our study included the following steps. First, we determined that **1** and **2** differentially changed the inhibitory profiles of individual enzymes within the DYRK and CLK families. Keeping this information in mind, we turned to theoretical calculations to understand how one single substitution can reshape the target selection. Methodologically, we performed quantum chemical calculations of the electronic parameters of **1** and **2,** followed by molecular docking and molecular dynamics (MD) simulations of complexes formed by each compound with active sites of key targeted kinases. Our sequential analysis unveiled non-trivial peculiarities of complex formation by **1** and **2** with their preferred DYRK/CLK targets. Next, we asked whether the calculated peculiarities were translatable into RNA splicing regulation and cytotoxicity in patient-derived glioblastoma multiforme (GBM) neurospheres. Omics data demonstrated numerous consequences of a **2**-CLK4 interaction on the expression of individual genes; these alterations were concomitant with GBM cell death. Overall, the differential donor-acceptor nature of the substituent is a critical prerequisite for the rational design of pyrido[3,4-*g*]quinazoline-based splicing antagonists. 

## 2. Materials and Methods

### 2.1. Pyrido[3,4-g]quinazoline Derivatives 

Compounds **1** and **2** (100 mg scale) were synthesized from commercially available 2-chloro-4-nitrobenzoic acid using a multistep procedure [11]. Compound **2** was obtained in 11 steps with an overall 8% yield, whereas **1** was prepared in 12 steps (6% yield).

### 2.2. In Vitro Kinase Assays

The inhibitory potencies of compounds **1** and **2** against DYRK1-4 and CLK1-3 were determined as reported by Rusina et al. [12]. The reaction mixture contained 0.1 mg/mL polyE4Y, the purified enzyme (0.1–0.7 pM), ^33^P ATP (1 μM) and the tested or reference (staurosporine) compounds at different concentrations. Mixtures were incubated for 2 h at 30 °C and spotted onto the ion exchange filter. The unbound phosphate was removed by extensive washing of filters in phosphoric acid. The IC_50_ and standard error (SE) values were obtained using GraphPad Prism 8 by log (inhibitor) vs. response (three parameters) curve analysis [13].

### 2.3. Quantum Chemical Calculations

Quantum chemical calculations were performed using a GAUSSIAN09 rev.C.01 [14]. For optimization of geometrical parameters and the solution of the oscillation problem, a DFT method, M06-2X [15] with a TZVP basis set [16,17], was used. The geometrical parameters of the N-1 (cation radical) and N+1 (anion radical) systems were optimized for the evaluation of electron parameters such as ionization potential IP and affinity to electron AE. Values IP and AE were calculated as the difference between enthalpies of the neutral close-shell and open-shell atomic/molecular systems [18] using the following formulas:*IP* = *E*(*A*^+^) − *E*(*A*),(1)
*IP* = *E*(*A*^+^) − *E*(*A*),(2)

Values of the highest occupied molecular orbital (HOMO), the lowest vacant molecular orbital (LUMO) [18] and the HOMO-LUMO gap values were calculated as the difference between AE and IP:(3)$$GAP_{HOMO-LUMO}=EA-IP,\;\mathrm{where}$$
$$IP=-\varepsilon_{HOMO}$$
$$EA=-\varepsilon_{LUMO}$$

The rigidity *η* of the molecule (a measure of its resistance to the charge transfer), the softness *S* (reciprocal to *η*), the electric negativity *χ* (a measure of the affinity of the molecule to electrons) and the electrophilicity index *ω* were calculated using the following formulas: (4)$$\eta =\frac{IP-EA}{2}$$
(5)$$S=\frac{1}{2\eta }$$
(6)$$\chi =\frac{IP+EA}{2}$$
(7)$$\omega =\frac{\chi^{2}}{2\eta }$$

### 2.4. Molecular Docking 

The geometrical parameters of target protein kinases CLK4 and DYRK3 in complexes with ligands were obtained from the Protein Data Bank [19] as well as PDB codes 6FYV [20] and 5Y86 [21]. Model structures of proteins were prepared using Schrödinger’s Protein Preparation Wizard. The H atoms were minimized, and the side chains of amino acid residues were added. Bond multiplicities were restored, and the solvent molecules were removed. The structures were limitedly optimized in the OPLS4 force field.

We applied a protocol of induced-fit docking (IFD) with the following conditions: the protein and the ligand were flexible, the grid template was 15 Å and the limited optimization of the amino acid residues was around 5 Å from the ligand. Docking solutions were ranked based on the docking score, the ligand efficiency (a parameter that considers atom-to-atom distribution of the scoring function) and а model value of energy (E_model_), which includes the docking score value, the energy of unbound interactions and energies of ligand positioning in the binding site. Furthermore, the free energies of binding to the target were calculated for the preferred ligand positions (ΔG_MM-GBSA_) using the generalized Born model with variable dielectric permeability. Water as a solvent was considered implicitly. 

In search of tentative binding sites for **1** and **2**, we analyzed the published X-ray crystal structures of the DYRK3 and CLK4 complexes with the respective inhibitors 5-[(3-chlorophenyl)amino]benzo[c][2,6]naphthyridin-8-carbonic acid (PDB ligand code **3NG**) and 7-methoxy-1-methyl-9H-β-carboline (PDB ligand code **HRM**) [20,21]. 

### 2.5. MD Simulations

Complexes of **1** and **2** with the respective target protein kinases were placed into an orthorhombic system with a buffer zone of 26 Å from the protein surface. The system was filled with 0.15 М NaCl in water. The solvent model was TIP3P, and the surrounding area was NPT. Multistage equilibration with heating and restrained minimization of the solvent and solute was used. The following options were selected for the MD calculations: RESPA integrator 2.0 fs, Nose–Hoover chain ensemble and a 9.0 Å cutoff for long-range electrostatics. The period of the recorded dynamics simulation was 100 ns at a temperature of 310 K (37 °C). The system preparation included preliminary optimization and equilibration of components. 

### 2.6. Analysis of MD Trajectories

The MD simulation trajectory was recorded in 5000 frames. The simulation analysis included an assessment of the root-mean-square deviation (RMSD), root-mean-square fluctuation (RMSF) and a description of the interaction of ligand atoms with amino acid residues of binding sites. The following intermolecular interactions were registered: hydrogen bridges at distances <2.5 Å and an angle of 120° in D (donor)—H—A (acceptor) combination or 90° in H—A (acceptor)—X combination. Hydrophobic contacts included π-π stacking, π-cation stacking (4.5 Å) and other non-specific interactions such as the contacts between hydrophobic and aromatic (or aliphatic) amino acid residues of the ligand (<3.6 Å). Ionic interactions (or salt bridges) were registered at 3.4 Å between the charged atoms. Water bridges represented the contacts of the ligand with amino acid residues through H2 molecules < 2.8 Å and with an angle of 110° in D (donor)—H—A (acceptor) combination or 90° in H—A (acceptor)—X combination. The MD trajectory clustering procedure was carried out with a frequency of 10. The binding energy (ΔG_MM-GBSA_) of the ligand and protein was estimated for a statistically significant complex, i.e., the complex most often detected in the analyzed frames.

### 2.7. Cell Culture and Viral Transduction 

Patient-derived GBM cell lines were obtained as described [22]. The material was the pieces of surgical GBM specimens routinely excised for morphological examination. Surgical decisions were based solely on clinical grounds. Biological material was processed in the research laboratories after sample de-identification. The study was performed according to the principles proclaimed by the Helsinki declaration and was approved by the local Medical Ethical Committee (protocol #176 from 29 August 2019). GBM neurospheres were cultured in Dulbecco’s modified Eagle’s medium/F12 medium (Sigma, St. Louis, MI, USA) containing 2% MACS NeuroBrew-21 (Miltenyi Biotec, Bergisch Gladbach, Germany) supplement, 1% penicillin-streptomycin solution (Thermo Fisher, Waltham, MA, USA), 20 ng/mL basic fibroblast growth factor (bFGF, Sigma-Aldrich, St. Louis, USA) and 20 ng/mL epidermal growth factor (EGF, Sigma) at 37 °C, 5% CO_2_ in a humidified atmosphere. bFGF and EGF were added twice a week. The medium was changed every 5–10 days. Neurospheres were dissociated using StemPro Accutase (Thermo Fisher). Phoenix-GP cells (American Type Culture Collection, Manassas, VA, USA; CRL-3215) were cultured in DMEM-F12 medium containing 10% fetal bovine serum (Thermo Fisher), 1% penicillin-streptomycin solution and 1 mM sodium pyruvate (Thermo Fisher). Cells were passaged 2–3 times per week until reaching 80–90% confluency. The identity of GBM neursphere lines was confirmed by STR analysis [22]. Mycoplasma contamination was tested with LookOut Mycoplasma PCR Detection Kit. 

Lentiviruses were produced as described [23]. Briefly, Phoenix-GP cells were co-transfected with pCDH-EF1-MCS-IRES-Puro (System Biosciences)-based lentiviral vectors and two packaging plasmids psPAX2 and pMD2.G (Addgene, Watertown, USA). On the next day, the medium was changed. Lentivirus-containing supernatants were harvested 72 h later and filtered through a 0.45 μm syringe filter. On the day of transduction, GBM spheres were dissociated into single cells with StemPro Accutase and incubated with lentivirus-containing supernatants for 24 h in the presence of 8 μg/mL polybrene (EMD Millipore, Burlington, VT, USA). Two days after the infection, the transduced cells were selected with 1 mg/mL puromycin (Sigma) for 3 days.

### 2.8. Cell Viability Assays 

GBM neurospheres were dissociated using StemPro Accutase (Thermo Fisher) and counted on a Countess II automated cell counter (Thermo Fisher) with Trypan blue reagent to exclude non-viable cells (Thermo Fisher). Cells were plated into 96-well plates (6 × 10^3^/well) overnight. Compounds **1** and **2** were added to each well, and cells were incubated at 37 °C, 5% CO_2_ for 5 days. Cell viability was assessed with AlamarBlue reagent (Thermo Fisher) by measuring the fluorescence on a Fusion α-FP HT universal microplate analyzer (Perkin-Elmer, Waltham, MA, USA) with the excitation filter for 535 nm and the emission filter for 620 nm. 

### 2.9. Precipitation of CLK4 Interacting Proteins

GBM cells overexpressing Fc-CLK4 or Fc-tagged control protein were incubated on ice for 30 min with the lysis buffer (20 mM Tris pH 7.5, 150 mM KCl, 10 MgCl_2_, 0.5 mM dithiotreitol (DTT), 0.5% NP-40, 0.1% sodium deoxycholate, and the protease inhibitor cocktail) and centrifuged for 15 min, 20,000× *g* at 4 °C. The cleared lysate was used for precipitation. Thirty μL of the Protein A/G magnetic beads (Thermo Fisher) were incubated with GBM cell lysate for 1 h at room temperature under constant agitation. Compound **2** was added at a final concentration of 10 µM. After incubation, the beads were washed once with the lysis buffer and thrice with phosphate-buffered saline. The bound proteins were eluted with the buffer (8 M urea, 2 M thiourea, 10 mM Tris-HCl pH 8) and subjected to LC-MS/MS analysis.

### 2.10. RNA Isolation and RT-PCR 

RNA was isolated using an RNeasy mini kit (Qiagen, Hilden, Germany). The RNA concentration was determined on a Nanodrop One C Spectrophotometer (Thermo Fisher). cDNA was synthesized using Maxima H Minus cDNA Synthesis Master Mix (Thermo Fisher) according to the manufacturer’s protocol. PCR was performed on a ProFlex PCR System (Thermo Fisher) with 5X ScreenMix-HS reagent (Evrogen, Moscow, Russia). Cycling conditions were 95 °C for 150 s and then 34 cycles at 95 °C for 30 s, 60 °C for 30 s and 72 °C for 30 s. Amplified DNA was analyzed by agarose gel electrophoresis. Primer sequences were as follows: UPP1_for: GAA AAC GGA CCT TAA CAA GAA GC; UPP1_rev: GAT ACG CCT GCT TGT CCT TCT; MCL-1_for: CTC GGT ACC TTC GGG AGC AGG C; MCL-1_rev: CCA GCA GCA CAT TCC TGA TGC C; RIOK3_for: CCA GTG ACC TTA TGC TGG CTC AGA T; RIOK3_rev: GGT CTG TAG GGA TCA TCA CGA GTA; CCNA2_for: TCC TCG TGA CTG GTT AGT T; CCNA2_rev: CCC GTG ACT GTG TAG AGT GC; RPL22L1_for: ATG GCG CCG CAG AAA GAC; RPL22L1_rev: TGC CTA GTC CTC CGA CTC TGA TT.

### 2.11. Peptide Preparation for LC-MS/MS Analysis 

Eluates from immunoprecipitation reactions were incubated with DTT up to the final concentration of 5 mM at room temperature for 40 min. Then proteins were alkylated with iodoacetamide (up to the final concentration of 10 mM) at room temperature for 20 min in the dark. Alkylated samples were diluted by the addition of 50 mM ammonium bicarbonate solution at a ratio of 1:4. Trypsin (1:100 *w*/*w*) was added, and samples were incubated at 37 °C for 14 h. The reaction was stopped by the addition of formic acid up to a final concentration of 5%. The tryptic peptides were desalted using an SDB-RPS membrane, vacuum-dried and stored at −80 °C before LC-MS/MS analysis. Prior to LC-MS/MS analysis, the samples were re-dissolved in 5% acetonitrile with 0.1% trifluoroacetic acid solution and sonicated. 

### 2.12. LC-MS/MS Analysis 

The proteomic analysis was performed on a Q Exactive HF mass spectrometer. Samples were loaded onto 50 cm columns packed in-house with C18 3 μM Acclaim PepMap 100 (Thermo Fisher), with an Ultimate 3000 Nano LC System (Thermo Fisher) coupled to the MS (Q Exactive HF, Thermo Fisher). Peptides were loaded onto the thermostatically controlled column at 40 °C in buffer A (0.2% formic acid) and eluted with a linear (120 min) gradient of 4% to 55% buffer B (0.1% formic acid, 80% acetonitrile) in A at a flow rate of 350 nL/min. Mass spectrometry data were stored during automatic switching between MS1 scans and up to 15 MS/MS scans (a topN method). The target value for MS1 scanning was set to 3∙106 in the range of 300–1200 *m*/*z* with a maximum ion injection time of 60 ms and a resolution of 60,000. Precursor ions were isolated at a window width of 1.4 *m*/*z* and a fixed first mass of 100.0 *m*/*z*. Precursor ions were fragmented by high-energy dissociation in a C-trap with a normalized collision energy of 28 eV. MS/MS scans were saved with a resolution of 15,000 at 400 *m*/*z* and at a value of 1 × 10^5^ for target ions in the range of 200–2000 *m*/*z* with a maximum ion injection time of 30 ms.

### 2.13. Protein Identification and Quantification

Raw LC-MS/MS data from the Q Exactive HF mass spectrometer were converted to .mgf peaklists with MSConvert (version 3). For this procedure, we used the following parameters: “—mgf—filter peakPicking true” [1,2]. For thorough protein identification in samples from immunoprecipitations, the generated peak lists were searched with MASCOT (version 2.5.1) and X! Tandem (ALANINE, 2017.02.01) search engines against UniProt human protein knowledgebase with the concatenated reverse decoy dataset. The precursor and fragment mass tolerance were set at 20 ppm and 0.04 Da, respectively. Database-searching parameters included tryptic digestion with one possible missed cleavage, static modification for carbamidomethyl (C) and dynamic/flexible modifications for oxidation (M). For X! Tandem, we also selected parameters that allowed a quick check for protein N-terminal residue acetylation, peptide N-terminal glutamine ammonia loss or peptide N-terminal glutamic acid water loss. The result files were submitted to Scaffold 4 software (version 4.0.7) for validation and meta-analysis. We used the local false discovery rate scoring algorithm with standard experiment-wide protein grouping. For the evaluation of peptide and protein hits, a false discovery rate of 5% was selected. False positive identifications were based on reverse database analysis. We also set protein annotation preferences in Scaffold to highlight Swiss-Prot accessions among others in protein groups. Functional analysis of differentially precipitated proteins was performed using the STRING service [24]. 

### 2.14. Plasmid Construction

DNA fragments encoding CLK4 were amplified from cDNA by PCR technique using primers EcoRI-CLK4 (5′-AAAAGAATTCCGGCATTCCAAAAGAACTCAC-3′)/CLK4-BamHI (5′-ATAAGGATCCAGTAAGACCACTGATTCCCATTTC-3′) and cloned into EcoRI/BamHI sites of the pCDH-Fc vector (22). The resulting plasmid was named pCDH-Fc-CLK4. The absence of unwanted mutations in the inserts and vector-insert boundaries was verified by Sanger sequencing.

### 2.15. RNA Sequencing 

cDNA libraries for paired-end sequencing were prepared using TruSeq Stranded mRNA-Seq Library Preparation Kit (Illumina, San Diego, CA, USA) according to the manufacturer’s protocol. Samples were sequenced with an Illumina HiSeq 2500 system; 125 bp paired-end reads were generated. 

### 2.16. Analysis of Gene Expression and Alternative Splicing 

The RNAseq reads were trimmed for quality, and the paired reads were cropped using Trimmomatic (v. 0.35). Trimmed RNAseq reads were quantified against *Homo sapiens* GRCh38 genome annotation at the transcript level using Salmon (v.1.4). The results were aggregated to the gene level using the R package tximport. Differentially expressed genes were identified using the R package DESeq2. To compare samples from different studies, we conducted surrogate variable analysis with ComBat using the R package sva to correct data from intra-study batch effects. Principal component analysis (PCA) was used to compare differences between all samples using the FactoMineR package in R. GSVA enrichment analysis was performed using the GSVA package in R against the hallmark, ontology and canonical pathways gene sets from the Molecular Signatures Database (MSigDB). The GSVA function in the GSVA package returns an enrichment score for each sample for the given signature. The *t*-test was used to compare the enrichment score between groups for the given signature. 

For alternative splicing analysis, the trimmed RNAseq reads were mapped to the *Homo sapiens* GRCh38 genome annotation using STAR. The STAR software (v. 2.7.9) was used in a 2-pass mode, where the first pass was used to identify non-annotated junctions in the input data. The genome index was generated using annotated junctions and the common set of junctions found during the first-pass mode. During the second pass, we re-ran all samples with this genome. To compare splicing events between samples, we used the rMATS splicing tool with the parameters recommended by the developers: −c 0.0001 and -novel SS1 (multi-align reads were ignored as a default). Minor splicing differences were filtered out by thresholds of FDR < 0.05 and the IncLevelDifference value > 5%. Both annotated and unannotated splicing events were analyzed.

### 2.17. Statistics

All data are presented as mean ± SD. The number of replicates for each experiment was indicated in figure legends and referred to as independent biological replicates. Statistical differences between groups were evaluated using the unpaired two-tailed *t*-test unless specified otherwise. *p* < 0.05 was considered statistically significant.

## 3. Results

### 3.1. Differential Inhibition of DYRK/CLK Family Members by **1** and **2**

Compound **2** previously demonstrated promising therapeutic efficacy in the orthotopic model of transplanted GBM (compound FG1059 in [22]). This property has been attributed to the ability of **2** to interfere with mRNA splicing. Intriguingly, compound **1**, which differed from **2** solely in a single substitution at position 10 (Figure 1), did not alter RNA splicing (compound EY404 in [22]). These data suggested that **1** and **2** evoked differential effects on DYRK and CLK protein kinases, the splicing regulatory enzymes druggable by pyrido[3,4-*g*]quinazoline derivatives. To test this assumption and to identify the target(s) for the therapeutically promising compound **2**, we tested **1** and **2** against a panel of purified DYRK and CLK enzymes in in vitro kinase assays (Appendix A). Table 1 shows the differential sensitivity of individual protein kinases: compound **1** has >2-fold higher potency against DYRK1A, -1B, DYRK2 and CLK1 whereas **2** has 1000 and 10 times lower IC_50_ for CLK4 and DYRK3, respectively.

### 3.2. Electron Parameters of Compounds **1** and **2**

To characterize the differential reaction capabilities of **1** and **2**, we calculated the ionization potential (*IP*), electron affinity (*EA*), HOMO-LUMO gap and their derivatives, such as the rigidity (*η*), softness *(S*), electric negativity (*χ*) and electrophilicity (*ω*). Appendix A shows that the *IP* value for 1 was 1.4 eV smaller than for **2**. Also, EA and HOMO-LUMO gap values were lower for **1** compared to **2**. The molecular electrostatic potential (ESP) characterizes the regions involved in intermolecular interactions. In ESP maps of compounds **1** and **2** (Appendix A), the zones of H-bond acceptors (or electron density donors) are rendered in blue, whereas the H-bond donors (or electron density acceptors) are shown in red. The electron density of the amino groups in the pyrimidine ring of **1** was re-distributed, whereas in **2** the ESP value increased due to the shift of the electron density to -NO_2_ moiety. Thus, quantum chemical calculations revealed significant differences in donor-acceptor characteristics of **1** vs. **2**. One may hypothesize that these differences are attributable to the differential kinase inhibitory potency. 

### 3.3. Molecular Docking of **1** and **2** into DYRK3 and CLK4 Models

Next, we performed molecular docking studies to reveal the modes of interaction of **1** and **2** with DYRK3 and CLK4. Compounds **HRM** and **3NG** were used as reference ligands on the basis of published X-ray structures of complexes with DYRK3 and CLK4, respectively. In the complex with DYRK3, compound **HRM** forms hydrophobic interactions with Ile215, Phe220, Val223, Ala236, Leu290, Leu291, Val353 and Leu342. An additional binding to the target is achievable with Lys238, Glu253 and Asp355 via H-bonds (9). In complexes with CLK4, the benzonaphthyridine moiety of **3NG** is positioned between N-terminal Ala189, Val175 and C-terminal Leu295 and Val324 residues (10). Also, the aromatic moieties in **3NG** interacted with Phe241. The N2 atom of napthyridine, formed an H-bond between N H and Leu244 (2.9 Å). Compound **3NG** strongly interacted with Lys191 via a salt bridge (2.8 Å). The geometric parameters of **3NG** and **HRM** obtained during the molecular re-docking procedure reproduced the positions in the binding sites determined by X-ray diffraction analysis (Appendix A), indicating that the molecular docking protocol was correct. These data served as starting material for generating the *in silico* models of complexes **1** and **2** with DYRK3 and CLK4.

Figure 2 and Appendix A illustrate the interactions of **1**, **2** and reference compounds **HRM** and **3NG** with DYRK3 or CLK4 models. These figures and Appendix A showed a larger number of clash interactions (steric hindrance of ligand binding) with DYRK3 than with CLK4. Also, more H-bonds and amino acid residues close to -NH_2_ or -NO_2_ groups were detectable for DYRK3. Overall, **2** (-NO_2_) was more affine than **1** to both CLK4 and DYRK3. These results corroborated the ESP values for **1** and **2**: more extremums on the van der Waals surface would allow for more contacts of **2** with the target kinase. 

The calculated results of molecular docking (Appendix A) demonstrated that compound **2** was more affine to DYRK3 than **1** (differential IFD score 4.5 kсal/mol). Values E_model_ for **2** were lower than for **1**; this difference strongly suggests a better positioning of **2** in DYRK3. The reference compound **HRM** formed H-bonds with Lys238, whereas **1** and **2** formed H-bonds with Leu291 but not with Lys238. Compound **2** formed an H-bond with Asp355. The value ΔG_mm-bgsa_ for complexes **2**-DYRK3 was smaller than for **1**-DYRK3. The affinities of **1** and **2** to CLK4 were similar. The difference between the IFD scores was 0.3 kсal/mol. The E_model_ value for **2** was somewhat lower than for **1**, suggesting a preferred positioning of **2** in the binding site. H-bonds between Leu244 and **1**,**2** were similar to the reference ligand. The H-bonding with Lys191 or Asp325, detectable for **3NG**, was observed for neither **1** nor **2**. The energy of binding ΔG_mm-GBSA_ for **2**-CLK4 was lower than for **1**-CLK4, suggesting a bigger affinity of **2** to CLK4. Overall, the results of molecular docking provided evidence in favor of a more pronounced efficacy of binding of **2** to both DYRK3 and CLK4. We next used MD to analyze the behavior of **1** and **2** in DYRK3 and CLK4 binding sites.

### 3.4. Analysis of Ligand–Target Interactions by MD

The best docking positions were selected for generating the starting systems for MD simulations. The positions of **1** and **2** as well as the frequencies of intermolecular contacts between the ligands and amino acid residues in the binding site were monitored for up to 100 nsec. The secondary structures of target protein kinases showed that CLK4 is rather compact, whereas DYRK3 contains free fragments: a loop between two β-sheets and a loop that follows the α-helix (Appendix A, circled). One may suggest that these fragments can undergo significant fluctuations during the MD procedure.

Analysis of RMSF graphs (Appendix A) indicates that the most mobile regions are loops 299–321 in CLK4 and terminal loops in DYRK3. These regions are distant from the ligand binding site. 

The evolution of RMSD over the time of the simulation is indicative of system equilibration. Fluctuations within 1–3 Å are normally acceptable for big globular proteins. For systems **1**-CLK4 and **2**-CLK4, the amplitude of protein fluctuations was ≤2 Å (gray curves in Appendix A, top panel). The dark red curves characterize the ligand equilibration relative to the protein. In the **1**-CLK4 complexes, the ligand movements in the binding site are relatively free; RMSD is bigger than the amplitude of fluctuations. In contrast, in the **2**-CLK4 system, the ligand is fixed in the binding site. 

Analyzing RMSD fluctuations for DYRK3, we detected periods of strong disturbances, such as events around 80 nsec for **1**-DYRK3 and 30 nsec for **2**-DYRK3 (Appendix A, bottom panel). One reason for these fluctuations could be the above-mentioned differences in the secondary structures of target proteins (Appendix A). Compounds **1** and **2** formed hydrophobic contacts, hydrogen bridges and water-mediated links in the DYRK3 binding site. As with CLK4, compound **2** (-NO_2_) formed ionic bonds (or salt bridges) with neighboring residues. Each ligand generated a hydrogen bridge that remained persistent throughout the entire period of the simulations (Figure 3). Similar to CLK4, the -NH_2_ group of the aromatic ring in **1** was not involved in interactions with amino acid residues. In contrast, the nitro moiety in **2** formed a water-mediated contact with Asp294. Thus, compounds **1** and **2** demonstrated differential positioning and intermolecular interactions with amino acid residues in the binding sites of DYRK3 and CLK4. Most importantly, the -NH_2_ group in **1** was not involved in interactions with neighboring residues, whereas the -NO_2_ moiety in **2** participated in the bonding. Altogether, MD procedures indicated that a single group substitution at position 10 is critical for conferring the ability to form complexes with individual protein kinases. One might expect that the reshaping of target selection would be mechanistically relevant to the differential effects of **2** vs. **1** on RNA splicing. 

Next, we analyzed the frequencies and types of intermolecular interactions (that is, H-bonds, hydrophobic contacts, salt or ionic bridges and water-mediated interactions) between ligands and the amino acid residues in the binding sites. Histograms in Figure 3 show an increase in contacts with CLK4 in the course of simulations. For instance, the value 0.25, characteristic for the contact between **1** and Leu167, surmises that this contact is preserved within 25% of the time of the simulation. Values > 1 indicate that the amino acid residues are capable of forming multiple contacts (e.g., two or more H-bonds between the ligand and the residue). Our results demonstrated that each ligand established prolonged interactions with amino acid residues 167–325, although the details of interaction modes differed. Compound **1** formed more H-bonds than **2**, whereas the frequencies of the hydrophobic and water-mediated interactions were similar. Most importantly, **2** and CLK4 formed ionic interactions for as long as 60% of the time of the simulations. In the **1**-CLK4 system, these interactions were not detectable. We attributed this finding to differential donor-acceptor properties of **1** (-NH_2_) vs. **2** (-NO_2_). Figure 3 shows that the -NO_2_ moiety is involved in contact with surrounding amino acid residues in the binding site. This fact is explained by the acceptor characteristics of **2**, such as π and cation stacking interactions with Phe241 and a salt bridge with Asp325. 

The clustering of MD trajectories revealed the quantitative and qualitative differences of ligand–target interactions. Unlike **1**, compound **2** formed ionic bonds with Asp294 and Glu339 in DYRK3, and with Lys191 in CLK4. The same bonding was visualized with the reference CLK4 inhibitor in complexes resolved by X-ray (PDB ID 6FYV). Thus, the clustering procedure yielded statistically significant (i.e., the most frequently detected during MD simulations) geometrical parameters of complexes **1** and **2** with DYRK3 and CLK4. The analysis of differential electronic parameters and amino acid surroundings indicated that the kinase inhibitory potencies were associated with the positioning of each compound in its binding site.

We next studied the positioning of **1** and **2** after the clustering procedure vs. the respective reference DYRK3 and CLK4 inhibitors. The positions of **1**, **2** and the reference DYRK3 inhibitor **HRM** in the DYRK3 catalytic cleft coincided; that is, the pyridoquinazoline fragments of **1** or **2** and the pyridoindole scaffold of **HRM** were located identically (Figure 4, top panel). Ligands were surrounded with hydrophobic amino acid residues; the N^7^ atom adjacent to Lys238 formed H-bonds. The position of **HRM** supported the formation of the hydrogen bridge between N^2^ and Lys238 (Appendix A). The energy values of ligand–target complexes differed by 3.9 kcal/mol: the complex **2**-DYRK3 was more stable. These results demonstrated that **1** and **2** interacted with similar amino acid residues. The only substantial difference is the involvement of the -NO_2_ group in contact with negatively charged aspartate residues. The complex **2**-DYRK3 was energetically more stable than **1**-DYRK3, which correlated with the kinase inhibitory potencies (Table 1). 

In contrast to DYRK3, the positions of **1** and **2** in the active site of CLK4 varied (Figure 4, bottom panel). The N^3^ atom in **1** was surrounded by the same residues as N^2^ of the reference inhibitor **3NG**. Furthermore, the -NH_2_ group at position 10 of **1** was located in the same area as the -NH- linker of **3NG**. The pyridoquinazoline moiety of **1** was found in the same area as the tricyclic scaffold of **3NG**. The N^7^ atom of **1** was oriented toward Lys191, whereas the carboxy group in **3NG** formed a salt bridge with this residue (see also Appendix A). Importantly, the behavior of the **2** was different. The positions of N^7^ and N^1^ coincided with N^2^ and N^6^ of **3NG**, and the -NO_2_ moiety was exposed to Lys191 forming salt bridges. The energy parameters of ligand binding in the active site of CLK4 differed by almost twofold, indicating a bigger stability of the complex **2**-CLK4 compared to the stability of **1**-CLK4. This difference is in line with a greater CLK4 inhibitory potency of **2** vs**. 1** (Table 1). 

Altogether, our calculations suggested that differential kinase inhibitory potencies of **1** and **2** were associated with their positioning in the target and interactions therein. Our calculated values of the binding energy in ligand-kinase complexes correlated with experimental enzyme inhibitory potencies. Most importantly, in the DYRK3 catalytic cleft, **1** and **2** demonstrated similar dynamic behavior, whereas in CLK4, their positioning differed substantially. Thus, based on our calculations, one may expect that the previously reported differential effects of **1** and **2** [22] were attributable to the potency of CLK4 inhibition.

### 3.5. Differential Effect of Compounds 1 and 2 on RNA Splicing in GBM Neurospheres

To translate the results of in silico calculations and in vitro kinase inhibition assays to the cell-based systems, we used patient-derived GBM neurospheres as the antitumor potency of **2** that has been demonstrated in this experimental system [22]. Because the in silico parameters of ligand–target complex formation and DYRK/CLK inhibitory profiles differed for **1** and **2**, we first compared the effects of each compound on RNA splicing in patient-derived GBM cells. In doing so, we tested these compounds on the subset of five mRNAs known to be particularly sensitive to the small molecular weight splicing inhibitors [25,26]. As a positive control, we used pladienolide B, the potent commercially available splicing inhibitor. Figure 5A shows that compound **2** evoked a stronger effect on RNA splicing than **1**. The potency of **2** for some targets was similar or even superior compared to pladienolide B. Importantly, the patterns of mRNA splicing in response to **2** and pladienolide B were not identical, suggesting that the two compounds act through different molecular mechanisms: pladienolide B inhibits the ubiquitous core spliceosomal protein SF3B1, whereas **2** impairs functions of tissue-specific SRSF family splicing factors by abrogating their CLK/DYRK-dependent phosphorylation. Compound **2** was particularly potent in affecting the splicing of RPL22L1 pre-mRNA. This protein has been shown to play a key role in the regulation of the GBM cell phenotype [22]. Also, both **2** and pladienolide B altered the abundance of pre-mRNA of Mcl-1, an anti-apoptotic Bcl-2 family protein [27]. However, each compound acted in an individual manner; namely, pladienolide B affected the splicing of *MCL1* transcripts, whereas **2** decreased the steady-state pre-mRNA levels. The mechanism of the latter effect remains to be elucidated. One may hypothesize that **2** may promote the generation of Mcl-1 isoforms that undergo a nonsense-mediated decay [28] and are, therefore, undetectable by gel electrophoresis. Alternatively, **2** may target a yet-to-be-identified factor that decreases the *MCL1* gene transcription and/or accelerates the degradation of the transcripts. Regardless of the mechanism, this important effect of **2** should be deleterious for tumor cells in which Mcl-1 is critical for survival [27]. Interestingly, for CCNA2 RNA, we observed an apparent attenuation of the effect of **2** on the exon skipping along with the increase in the drug concentration. This result can be explained by the fact that, at the higher concentrations, **2** may affect the recognition of multiple splice sites in CCNA2 pre-RNA, leading to intron retention rather than exon skipping. The resulting mRNA will be undetectable by RT-PCR due to a longer length and/or lower stability. 

After evaluating the effects of **1** and **2** on RNA splicing, we tested whether these compounds affect the viability of GBM neurospheres derived from six patients (Figure 5B). Consistent with differential potencies in RNA splicing deregulation, compound **2** showed much stronger cytotoxicity than **1** for all tested cell lines. Of note, the smallest differences in the sensitivity to **1** and **2** were observed for 267 and 011 GBM cells, while the most pronounced differences were observed for 006 cells. To understand the reason underlying this discrepancy, we analyzed RNA sequencing data of the corresponding untreated GBM sphere lines (Figure 5C) and found that 006 cells expressed the highest level of CLK4 (a preferred target of **2**; Table 1) and the lowest level of DYRK2 (a preferred target of **1**; Table 1). The opposite trend was observed for 267 and 011 cells that expressed the lowest level of CLK4 and the highest level of DYRK2. This important result confirmed that the differential effects of **1** and **2** on GBM cells may be attributed to different potencies of these compounds against individual members of DYRK/CLK families.

### 3.6. Compound **2** Alters the Interactions of CLK4 with Its Substrates

Based on our quantum chemical calculations and in vitro kinase assays, we hypothesized that the stronger effect of **2**, compared to **1**, on RNA splicing may be explained by their differential inhibitory potency against CLK4. Therefore, we first tested if **2** can alter the interactome profile of CLK4 protein kinase. In so doing, CLK4 tagged with Fc fragment at the N-terminus (Fc-CLK4) or Fc-tagged Survivin (a CLK4 unrelated control) were exogenously expressed in 019 GBM neurospheres using lentiviral infection. Next, CLK4 and its interacting proteins were purified from cell lysates using G-protein-coated magnetic beads in the presence or absence of **2**. LC-MS/MS analysis of the proteins eluted from the magnetic beads demonstrated that, in the absence of **2**, CLK4 binding partners were enriched with splicing factors (false discovery rate, FDR = 3.44 × 10^−21^) and ribosomal proteins (FDR = 7.98 × 10^−30^) (Figure 6A and Appendix A). The latter group was rather unexpected. We tend to interpret this observation as a non-specific binding of ribosomal proteins to CLK4; otherwise, these interactions reflect yet unknown function(s) of this protein kinase. In vitro incubation with **2** significantly altered CLK4 binding partners (Figure 6B). Enrichment analysis of the proteins whose interaction with CLK4 was abrogated by **2** revealed the most substantial decrease in the proteins involved in RNA splicing (FDR = 1.6 × 10^−6^). This group includes serine/arginine splicing factors (SRSF), the major downstream targets of DYRK and CLK that play a critical role in GBM cell survival [29]. Consistently, **2** disrupted the complexes of CLK4 with SRSF3 and SRSF4 as well as with other splicing factors and spliceosome subunits U2AF2/U2AF65 and SNRNP70 involved in the recognition of 3′- and 5′-splicing sites, respectively (Figure 6C). Interestingly, proteins whose interaction with CLK4 was increased in response to **2** were enriched with ribosomal components (Figure 6B). This CLK4-ribosome binding may be attributed to altered kinase conformation by the inhibitor. Altogether, these data strongly suggested that compound **2** disrupted the interactions of CLK4 with spliceosomal proteins, which may subsequently affect RNA splicing. 

### 3.7. Effect of Compound **2** on the Transcriptome of GBM Cells

We next investigated whether compound **2** affects the transcriptome in GBM cells. The 019 GBM cells were treated with 3 µM of **2** for up to 24 h, followed by RNAseq analysis (Appendix A). Approximately 35,000 transcripts were identified and subjected to the principal component analysis (PCA), which demonstrated a good clustering of replicates and revealed significant differences between individual time points (Appendix A). 

The enrichment analysis of differentially expressed genes (*p*-value cutoff < 0.05 and fold change cutoff > 2) (Figure 7A,B) using the GO Biological Process database revealed a major up-regulation of DNA damage pathways, in particular, p53-related genes at 6 h and 24 h with **2**. Interestingly, by 6 h, compound **2** down-regulated the hypoxia-related genes, which may be related to the impaired functions of mitochondria. By 24 h, the cell adhesion pathway was substantially down-regulated, an observation attributable to the detachment of apoptotic cells from the substrate.

Finally, to understand the molecular mechanism underlying the effect of **2** on GBM cells, we analyzed its global effect on RNA splicing. Our results revealed >750 alternatively spliced transcripts by 6 h; these differences increased up to >1000 transcripts by 24 h (Figure 7C and Appendix A). Exon skipping was the most frequent type of splicing alteration induced by **2** in 019 cells (Figure 7D). The second one was the inclusion of mutually exclusive exons. The frequency of this event increased >2-fold by 24 h compared to the 6 h interval. Surprisingly, intron retention was the least frequent event suggesting that, in contrast to pladienolide B, compound **2** is unlikely to interfere with core subunits of the spliceosome. Rather, this **c**ompound can impair the splice site selection [25]. This observation supports the mechanistic role of **2** as a direct inhibitor of splicing protein kinases; the targeting of these kinases leads to hypophosphorylation of SRSF proteins, thereby impairing exon recognition. 

Finally, we performed the enrichment analysis of differentially spliced RNAs using the Reactome database (Figure 7E). Our results demonstrated that **2** largely affected the splicing of genes whose products regulate transcription, in particular, RNA polymerase II-associated events. These results are highly consistent with gene expression data (Figure 7A,B).

## 4. Discussion

The pyrido[3,4-*g*]quinazoline chemotype provides vast opportunities for the search for active compounds as DYRK/CLK antagonists. In particular, our group has shown the potency of pyrido[3,4-*g*]quinazoline derivatives in the deregulation of RNA splicing concomitantly with the antitumor efficacy in transplanted GMB xenografts [22]. In the present study, we combined the calculation chemistry and experimental approaches to address the mechanism whereby one structural unit can reshape the target selection and, therefore, the cellular effects. 

We demonstrated that a single group substitution at the periphery of the heterocyclic core, namely, -NO_2_ (**2**) instead of -NH_2_ (**1**) at position 10 of the pyrido[3,4-*g*]quinazoline scaffold, significantly changes DYRK/CLK inhibitory profiles. Compound **1** was more potent to DYRK1A, -1B, DYRK2 and CLK1, whereas DYRK3 and CLK4 were the preferred targets for compound **2**. May these key differences be associated with the reaction capabilities of **2** vs. **1**? The ESP maps showed a strong minimum at the nitro group of **2**. Furthermore, **2** had larger values of IP, the HOMO-LUMO energy gap and electrophilicity. However, regardless of these differences, **1** and **2** showed a similar in silico affinity to the catalytic domain of CLK4. Importantly, the IC_50_ values of CLK4 inhibition by **1** were three orders of magnitude higher than by **2,** whereas the energies of binding of each compound to this target differed only by 3 kcal/mol. One explanation of this difference is provided by MD simulations in which the amino group in **1** was virtually free from interactions with the neighboring amino acid residues, in contrast to the nitro group in **2**. Thus, the complex **2**-CLK4 is thermodynamically more stable than **1**-CLK4. Considering the DYRK3 target, the affinity of **2** to the enzyme’s catalytic site was bigger than in the case of **1**. Moreover, the energy of the binding of **2** to DYRK3 was lower than the respective value for **1**. MD analysis revealed that, unlike the amino group in **1**, the nitro group in **2** was involved in intermolecular interactions with the amino acid residues during the entire period of simulations. Thus, compound **2** is the preferred binder of CLK4 and DYRK3, the two enzymes that regulate RNA splicing. Importantly, -NO_2_ to -NH_2_ substitution did not reduce the affinity to DYRK/CLK family members. Rather, the specificity to individual isoenzymes was altered: the -NO_2_ moiety (**2**) favored the interaction with DYRK3 and CLK4 while -NH_2_ (**1**) increased the affinity toward DYRK1A and CLK1 as determined in in vitro kinase assays. 

Recently, a series of inhibitors with differential selectivity to individual members of DYRK/CLK families have been generated, including the compounds SM08502 and CTX-712 that entered clinical trials (NCT05084859, NCT03355066, NCT05732103, JapicCTI-184188). A number of molecules are in preclinical stages [30]. Compound SM08502 was the most efficacious against CLK2, CLK4, DYRK1A and DYRK1B; its effect on RNA splicing was mediated by reduced phosphorylation of the SRSF6 protein [31]. On the other hand, CTX-712 inhibited SRSF3/4/6 phosphorylation; however, the effect of this compound on the kinase activities of DYRK/CLK family members has not been disclosed [32]. The anilino-2-quinazoline derivative DB18 was remarkably potent against CLK1, CLK2 and CLK4 but showed negligible activity against DYRK1A [33]. Finally, Lee Walmsley et al. [34] reported the hit pyrrolopyrimidine compound with high selectivity for DYRK1A and DYRK1B compared to DYRKs2-4. Thus, DYRK/CLK inhibitors, including compounds **1** and **2**, have shown a certain degree of specificity toward the individual isoenzymes. Nevertheless, structural reasons underlying difficulties in the design of highly selective isoenzyme inhibitors make off-target effects on other DYRK/CLK family members hardly avoidable [35]. 

Clinically, SM08502 down-regulated CLK1 and impaired the alternative splicing in patients with advanced solid tumors. In some patients, tumor shrinkage and disease stabilization were achieved [36]. Treatment with CTX-712 caused splicing abnormalities in patients; partial (high-risk myelodysplastic syndromes, ovarian cancer) or complete (refractory acute myeloid leukemia) responses were observed in individual patients [37,38,39]. The DYRK/CLK inhibitor SM09419 has been shown to overcome the resistance to venetoclax [40]. These results demonstrate the high clinical relevance of DYRK/CLK inhibitors. However, the analysis of The Cancer Genome Atlas (TCGA) database shows that individual DYRK/CLK isoenzymes may play different roles (Appendix A). In gliomas, the expression of DYRK3 and CLK3 correlated with worse survival, while DYRK1A, DYRK2 and CLK1 were associated with better disease outcomes. On the other hand, in renal cancer, CLK2 strongly correlated with poor prognosis (*p* = 3 *×* 10^−8^), whereas DYRK1B was indicative of prolonged survival (*p* = 9 × 10^−4^). Based on these results and the differential cytotoxicity of **1** and **2** in patient-derived GBM cells (Figure 5B), one may expect that fine-tuning of the chemical structure would yield DYRK/CLK inhibitors with enhanced binding to the individual isoenzymes important in the specific tumor type.

## 5. Study Limitations

Regardless of the detailed in silico analysis, the X-ray structures of drug-target complexes are worthy for making an unambiguous judgment; also, **1** and **2** might have other splicing-related (and perhaps splicing-unrelated) targets. Indeed, the intracellular interactions of **2** are multifaceted: RNAseq studies identified genome-wide changes in the expression of multiple genes with critical functions for tumor cell biology, namely, the DNA damage response, p53 signaling and transcription. In the present study, these changes were translated into the cytotoxicity for patient-derived GBM neurospheres, demonstrating the practical applicability of the pyrido[3,4-*g*]quinazolines. It is the subject of future research to uncover how the pharmacological targeting of splicing is multiplied into a plethora of cytotoxic phenomena. On the one hand, such a manifold net effect is therapeutically beneficial for intractable tumors, including GBM. On the other hand, caution is needed when evaluating the clinical perspective: the fundamental role of RNA splicing may underlie the potential side effects. From this viewpoint, the differential potency of **1** and **2** due to a single substitution indicates that a pipeline of derivatives with the desired characteristics can be generated based on the pyrido[3,4-*g*]quinazoline scaffold. 

## 6. Conclusions

Protein kinases that phosphorylate spliceosomal proteins represent promising targets for antitumor therapy, including GBM. However, despite the structural similarity of their active sites, these enzymes perform different functions and may have a different impact on patients’ survival. Therefore, it is critical to discover the inhibitors specific to individual proteins rather than for all members of the corresponding kinase family. In this study, we demonstrated that the pyrido[3,4-*g*]quinazoline scaffold can serve as a source of compounds with differential inhibitory efficacy against individual DYRK and CLK enzymes. Inhibition of the splicing kinase CLK4 by compound **2** was a result of a single structural substitution in the pyridoquinazoline scaffold. The interaction of CLK4 and **2** was demonstrated by computational methods; its consequences were further confirmed by a number of high throughput techniques in cell-based assays. The net result of CLK4 inhibition by **2** is a widespread alteration of RNA splicing that led to numerous changes in gene transcription patterns. RNA splicing perturbations triggered by **2** were translated in the cytotoxic potency of this compound against a panel of patient-derived GBM neurospheres. These findings favor the possibility of the rational design of antitumor splicing antagonists based on the perspective chemotype. 

## Figures and Tables

**Figure 1 cancers-16-00834-f001:**
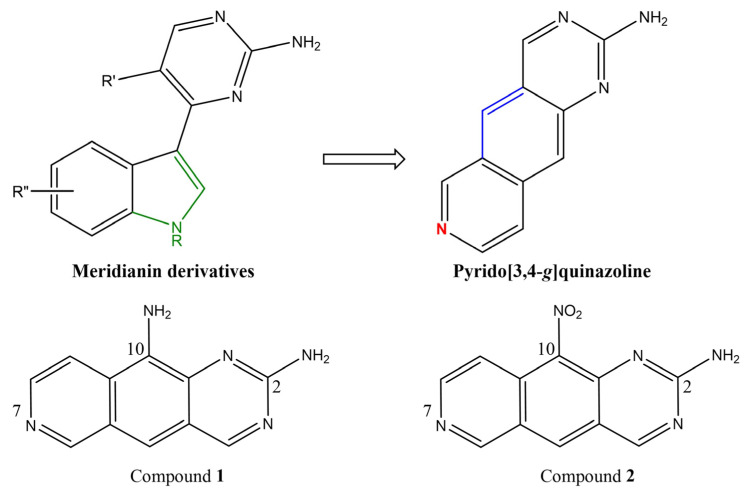
The pyrido[3,4-*g*]quinazoline scaffold of DYRK/CLK inhibitors (compounds **1** and **2**) investigated in this study.

**Figure 2 cancers-16-00834-f002:**
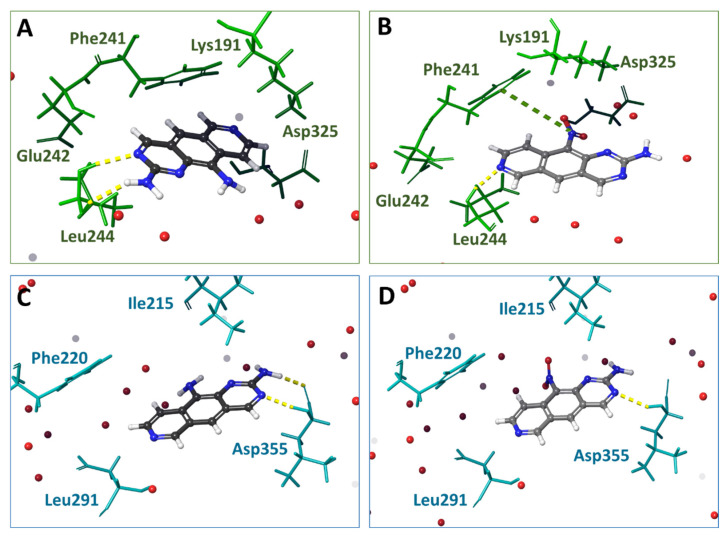
Docking positions of compounds **1** (**A**,**C**) and **2** (**B**,**D**) in the active site of CLK4 (**A**,**B**) and DYRK3 (**C**,**D**). H-bonds are shown as yellow dotted lines, and water molecules are represented as red balls.

**Figure 3 cancers-16-00834-f003:**
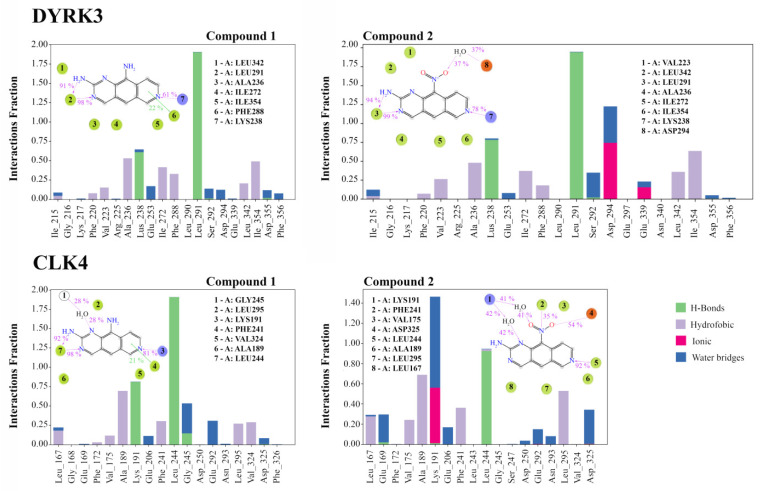
MD interactions of DYRK3 and CLK4 with compounds **1** and **2**. The following intermolecular interactions were registered: hydrogen bridges, hydrophobic contacts (π-π and π-cation stacking), contacts between hydrophobic and aromatic (or aliphatic) amino acid residues of the ligand as well as ionic interactions (or salt bridges) and water bridges. The value 1.0 corresponds to 100%; i.e., the contact is recorded during the entire simulation time. See Materials and Methods for details.

**Figure 4 cancers-16-00834-f004:**
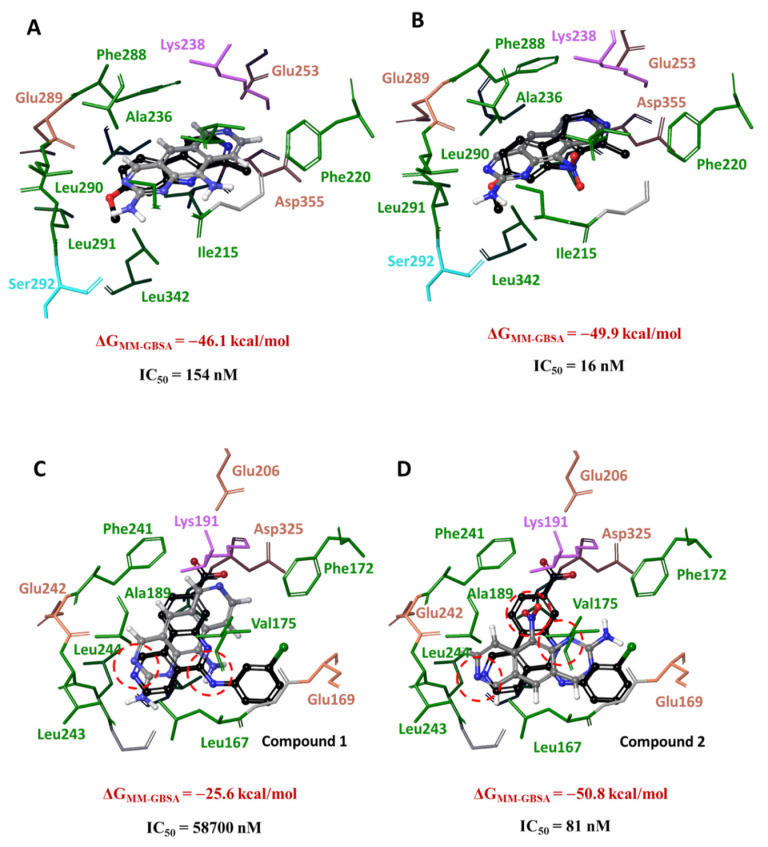
The positions of ligands in the catalytic clefts after clustering the MD trajectories. Top panel—DYRK3; bottom panel—CLK4. (**A**,**С**) Compound **1**; (**B**,**D**) compound **2**. Red dashed lines encircle the nitrogen-containing groups. Values IC_50_ were obtained in in vitro kinase assays (Table 1).

**Figure 5 cancers-16-00834-f005:**
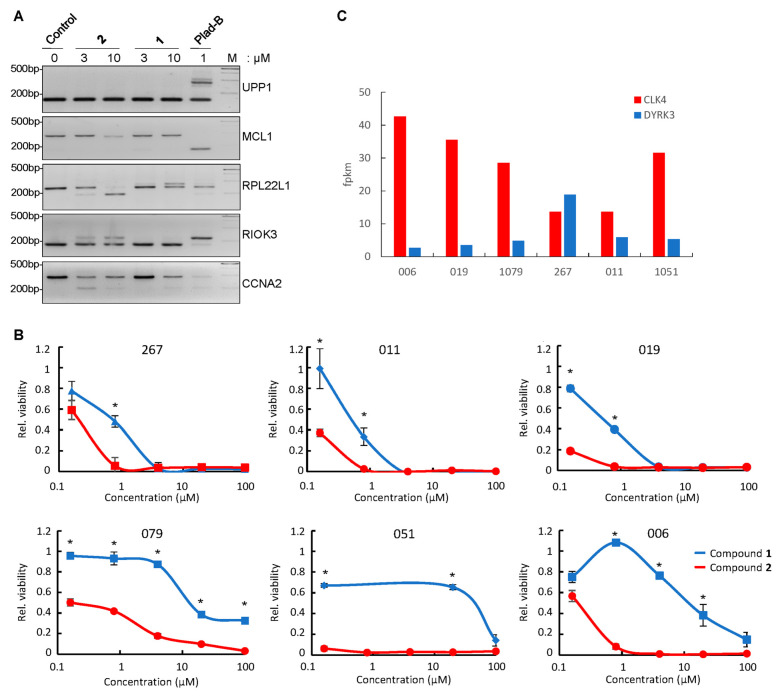
The effects of compound **2** on GBM cells. (**A**) RT-PCR analysis of splicing of UPP1, MCL1, RPL22L1, RIOK3 and CCNA2 RNAs in 019 GBM spheres either untreated (control) or treated for 24 h with indicated concentrations of **1**, **2** or pladienolide B (Plad-B); (**B**) Cell viability assays of GBM spheres obtained from n = 6 different patients (267, 011, 019, 079, 051, 006). Cells were treated with various concentrations of **1** (blue) or **2** (red) for 5 days (n = 6 biological replicates; * *p* < 0.01); (**C**) mRNA levels of CLK4 (red) and DYRK2 (blue) were determined by RNA sequencing analysis in GBM spheres obtained from n = 6 patients.

**Figure 6 cancers-16-00834-f006:**
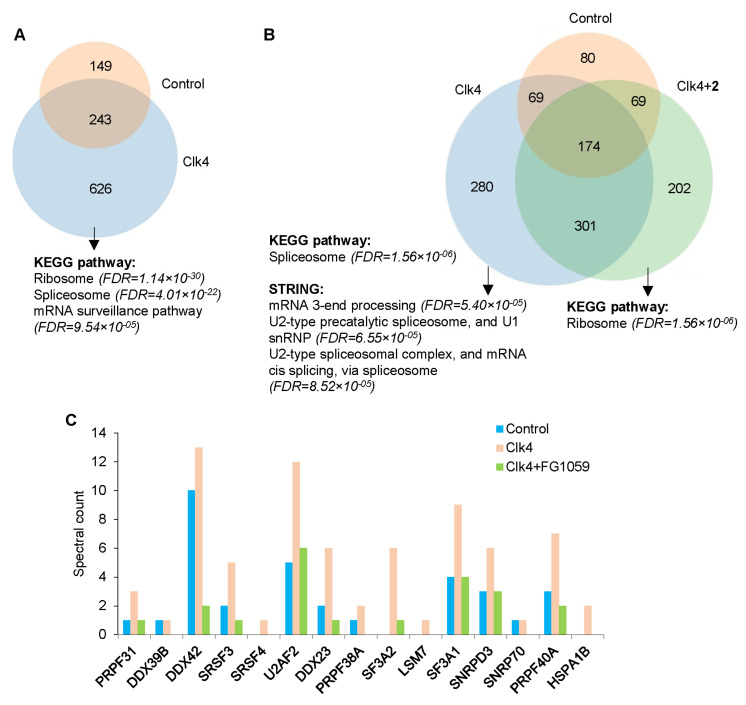
The CLK4 interactome in the presence or absence of compound **2**. (**A**) Venn diagram representing proteins that were co-purified with Fc-tagged CLK4 (blue) or control protein (red) and subsequently identified by LC-MS/MS. Proteins were considered differentially present if the spectral count fold change was ≥2. Differentially present proteins were subjected to enrichment analysis. The most significantly enriched terms and corresponding FDR values are indicated. (**B**) Venn diagram representing proteins that were co-purified with Fc-tagged CLK4 (blue), CLK4 in the presence of 10 μM **2** (green) or control protein (red) and subsequently identified by LC-MS/MS. (**C**) Histogram presenting the spectral count for spliceosomal proteins detected as in “(**B**)”.

**Figure 7 cancers-16-00834-f007:**
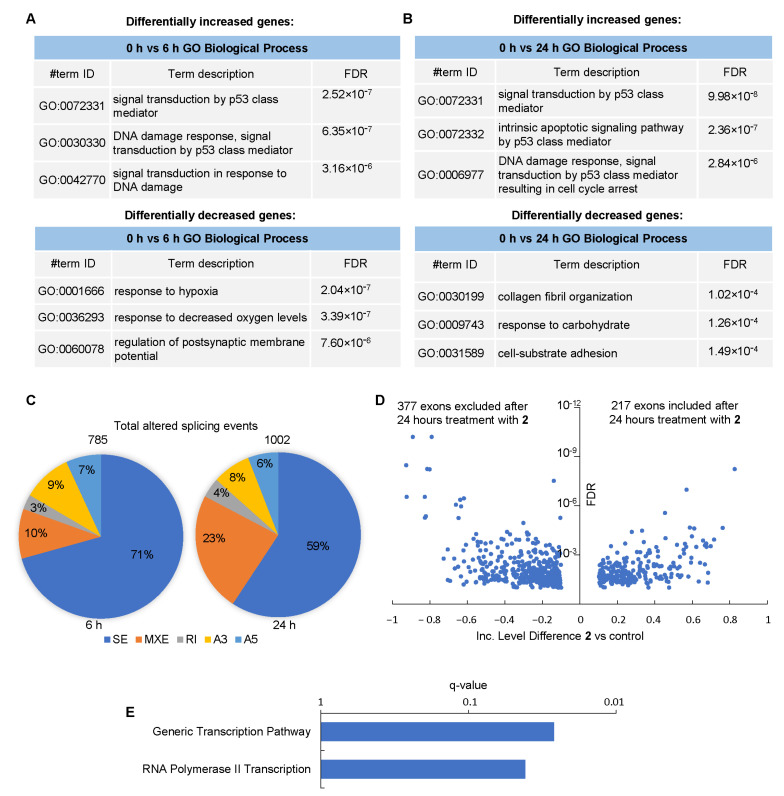
The effects of **2** on the transcriptome of GBM cells. (**A**,**B**) Enrichment analysis of genes differentially expressed (>2-fold differences) in 019 GBM cells after 6 h (**A**) and 24 h (**B**) of incubation with 3 μM of compound **2** compared to vehicle-treated (control) cells (n = 3 biological replicates). The GO Biological Process database was used to calculate the enrichment. (**C**) The ratios of the types of alternative splicing events observed after 6 h (left panel) and 24 h (right panel) with **2**. SE, skipped exon; MXE, mutually exclusive exons; RI, retention of intron. А3 and A5 are alternative 3′- and 5′-splice sites, respectively. (**D**) A volcano plot demonstrating significantly different alternative splicing events related to exon skipping after 24 h with **2**. (**E**) Enrichment analysis of alternatively spliced pre-mRNAs after treatment with 2.

**Table 1 cancers-16-00834-t001:** In vitro kinase inhibitory profiles of compounds **1** and **2**.

Protein Kinase	Compound 1	Compound 2
DYRK1A	28 ± 6	159 ± 18
DYRK1B	22 ± 5	63 ± 14
DYRK2	31 ± 6	80 ± 16
DYRK3	154 ± 18	16 ± 5
DYRK4	1220 ± 192	9420 ± 221
CLK1	41 ± 8	87 ± 12
CLK2	350 ± 28	391 ± 40
CLK3	4630 ± 150	4660 ± 122
CLK4	58,700 ± 1266	81 ± 10

The IC_50_ values are shown (mean ± SD of 3 measurements).

## Data Availability

The data presented in this study are available in the supporting Appendix A.

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
