# Peer review of "The Nitro Group Reshapes the Effects of Pyrido[3,4-*g*]quinazoline Derivatives on DYRK/CLK Activity and RNA Splicing in Glioblastoma Cells"

_cancers, 2024, doi:10.3390/cancers16040834_

Round 1

Reviewer 1 Report

Comments and Suggestions for Authors

This manuscript addressed a major problem in the development of DYRK/CLK inhibitors: the selectivity to individual family members. Through multiple approaches including quantum chemical calculations, molecular docking, and molecular dynamics simulations, they showed that a single chemical substitution in the pyrido[3,4-31 g]quinazoline scaffold changed the kinase inhibitory profile and yielded the splicing antagonist with a high cytotoxic potency against patient-derived GBM neurospheres.

This is an important research topic. The chemical and biophysical methods used are suitable and well carried out. The outcomes have the potential to shed new lights into the specificity and selectivity of DYRK/CLK inhibitors in GBM targeted therapy.

My major concerns are in the testing of compound 2 on RNA splicing and GBM neurosphere viability in Figure 5 (see detailed points below), and that Figure 6 and 7 did not include the necessary controls to provide some mechanistic explanation for why compound 2 shows much stronger cytotoxicity on GBM cells in Fig. 5.     

Major Points:

According to Fig. 5A, at 10 µM compound 2 evoked a much stronger effect on RNA splicing than 1, yet in Fig. 5B, the cytotoxicity of compound 1 and 2 at 10 µM is the same on GBM lines 011 and 267. Is it because compounds 1 and 2 elicit similar effects on RNA splicing in GBM 011 and 267 but very different effects in GBM 006, 051, 079? It will be much more informative if the same GBM lines are tested for RNA splicing in 5A and GBM viability in 5B.

What is a possible explanation for this GBM cell type-dependent response to 1 (Fig. 5B)? Is it possible that GBM 011 and 267 have high expression of common targets of 1 and 2 in the DYRK/CLK family whereas GBM 006, 051, and 079 are enriched for DYRK3/CLK1/CLK4 which are specific targets of 2? Some RT-PCR and Western blotting data may provide further insights.

Figure 6 tested the impact of compound 2 on CLK4 interactome. Given the difference in target sensitivity between 1 and 2, compound 1 would be a good control to show little or no impact on CLK4 interactions with substrates, which will enhance the statement on the selectivity of 2 on CLK4.

Figure 7 tested the effect of compound 2 on transcriptome. Again, it would provide a strong support to the overall conclusion of this paper if compound 1 were used as a control to show the differential impact of 1 and 2 on transcriptome.

Figure 7 A-B, Enrichment analysis of differentially expressed genes were conducted by grouping up-regulated genes and down-regulated genes. Given that changes in any certain biological processes may include both up-regulated and down-regulated genes, it can be argued to not separate the two pools and instead rank the biological processes based on differentially changed (including both ups and downs) genes.

Minor Points:

Page 7, Line 276-277, the statement “DYRK3 and CLK4 were the preferred targets for compound 2” left out CLK1 which has an IC50 that is very similar to that of CLK4.

Page 11, Fig. 5A, why compound 2 seems to have a stronger effect on slicing of CCNA2 at a lower concentration (3 vs 10 µM)?

Reviewer 2 Report

Comments and Suggestions for Authors

Two agents were screened to bind DYRK3 and CLK4 using molecular docking and MD simulation in this work. The authors are suggested to provide the RMSF plot for protein-ligand complexes. More bioactivities, including pharmacokinetics, would be needed for a work that could be published in Cancers.

Reviewer 3 Report

Comments and Suggestions for Authors

Congratulations for the excellent work and well-written article. I suggest just review some spelling English words.

1. The article is about the effects of Pyrido[3,4-g]-quinazoline derivatives in the pathway of DYRK/CLK protein kinase in the glioblastoma.

2. As well commented the topic is not original or new in the oncology; although, it is relevant in the context of brain tumors and to better understand the molecular basis of new compounds that can be in the future a medicine.

3. Add the use of this compound in the treatment landscape of brain tumors.

4. The methodology is clear and nothing needs to be improved.

5. As already indicate the arguments and conclusions address the mains topic.

Comments on the Quality of English Language

 I suggest just review some spelling English words.

Reviewer 4 Report

Comments and Suggestions for Authors

There are multitude of studies published in the area highlighted by the authors.

Comment #1: One of the primary concerns with this paper is the lack of clarity in defining the scope and objectives of the study. The authors have not clearly outlined the specific research questions or goals they aimed to address. As a result, the paper appears to be a collection of loosely connected ideas rather than a cohesive discussion.

Comment #2: Another major issue is the lack of rigorous methodology. The paper does not provide any information regarding the systematic studies used to perform this computation.

Comment #3: The paper also suffers from a lack of critical analysis and evaluation of the reviewed technologies. Merely describing and summarizing the existing technologies without offering any meaningful insights or comparisons diminishes the value of the work. The paper should have critically examined the strengths, limitations, and potential areas of improvement for each approach.

Comment #4: The overall organization and structure of the paper are inadequate. The flow of ideas is unclear, and there is a lack of coherence between sections. The paper should have presented a clear introduction, outlined the main themes or categories of technologies, and provided a concise summary or conclusion to tie the information together.

Comment #5: Spacing issues. Correct such issues throughout the manuscript.

Comment #6: Abbreviations must be explained next to the first mention or wherever appropriate to avoid reader struggle understanding the terminology. We have identified technical jargons which must be avoided by the authors.

Comment #7: Unrequired abbreviations otherwise should be discarded. 

Comments on the Quality of English Language

Thrust on the importance of key contributions. It is unclear from present version.

Round 2

Reviewer 1 Report

Comments and Suggestions for Authors

In this revised version, the authors have addressed my major concerns with further experimentation and clarification. 

One critical issue that still needs to be addressed before publication is the lack of indication of any statistical analysis on the new data in Fig. 5B and 5C, missing p value labels on the graphs and a statement on the statistical analysis in the figure legend. 

Author Response

Reply: We added indications of the p-values to the figure 5B and its’ legend. Figure 5C does not require statistical analysis as the bars just show fpkm values, a common measure of mRNA levels in RNA sequencing analysis. In this figure we do not compare one group vs the other, rather we just indicate which cell line have the highest and the lowest expression of the gene of interest.

Reviewer 4 Report

Comments and Suggestions for Authors

Physical interpretation of results is still missing and authors are advised to address in final upload to the journal

Comments on the Quality of English Language

Authors should proof-read their work and use less technical jargons

Author Response

1.Physical interpretation of results is still missing and authors are advised to address in final upload to the journal.

Reply: We added the paragraph on data interpretation on page 16 of the re-revised manuscript (colored in yellow).

“In the present study we found that inhibition of the splicing kinase CLK4 by compound 2 was a result of single structural substitution in the pyridoquinazoline scaffold. Interaction of CLK4 and 2 was demonstrated by computational methods; its consequences were further confirmed by a number of high throughput techniques in cell based assays. The net result of CLK4 inhibition by 2 is a widespread alteration of RNA splicing that led to numerous changes of gene transcription patterns. RNA splicing perturbations triggered by were translated in the cytotoxic potency of this compound against a panel of patient-derived GBM neurospheres. These findings favor the possibility of rational design of antitumor splicing antagonists based on the perspective chemotype.”

2.Authors should proof-read their work and use less technical jargons.

Reply: We double checked the terminology. Frankly, we are not sure we understand what the reviewer means under technical jargon. Indeed, many specific terms are currently being used by the developing languages of computational chemistry as well as omics methods. We will be grateful for indicating the awkward words; we are committed to polish the text.